



# Rate-Induced Transitions and Noise-Driven Resilience in Vegetation Pattern Dynamics

Lilian Vanderveken[1] and Michel Crucifix[1]

[1]Earth and Life Institute, Louvain-la-neuve, Belgium

**Correspondence:** Lilian Vanderveken (lilian.vanderveken@uclouvain.be)

**Abstract.** Understanding the resilience and stability of vegetation patterns under changing environmental conditions is crucial for predicting ecosystem responses to climate change. This study investigates the dynamics of vegetation patterns in response to a spatially homogeneous decrease in rainfall across the entire domain. Starting from high-rainfall with a stable homogeneous vegetated state, we applied various rates of rainfall reduction to observe system transitions. We find that rainfall decrease may
cause transitions to two or three pulse states, or abrupt shifts to bare soil depending on the rate of change, highlighting the significance of rate-induced tipping (R-tipping) in open dynamical systems.

We identified the pulse creation and destruction timescale ($\tau_{\text{pulse}}$) and the rearrangement timescale ($\tau_{rear}$) as the critical timescales which govern the system response to gradual environmental changes. The rearrangement timescale, significantly longer than $\tau_{\text{pulse}}$, is relevant for characterising the system behavior under slow perturbations. Dimensional analysis and sensi-
tivity analysis with numerical experiments further validate the fundamental connections between these timescales.

Additionally, we examined the impact of spatially and temporally structured noise on vegetation pattern resilience. Perturbations modeled as Gaussian stochastic processes with specific autocorrelation structures were applied to the system. We find that increased spatial autocorrelation in noise reduces pattern formation, while temporal autocorrelation at critical timescales significantly influences biomass mean and variance. The co-existence of multiple equilibria and unstable states, combined with
the presence of ghost attractors enhances system resilience by providing alternative stable configurations under fluctuating conditions.

These findings underscore the importance of considering slow timescales and structured noise in analyzing vegetation dynamics. Understanding these factors is essential for predicting ecosystem resilience and developing strategies to manage vegetation systems under climate variability.

## 1 Introduction

Understanding the dynamics of vegetation patterns under varying environmental conditions is crucial for predicting ecosystem responses to climate change. Vegetation patterns, such as the regular arrangement of plant patches, arise from complex inter-





actions between biological processes and environmental factors (Klausmeier, 1999a). These patterns are particularly sensitive

to changes in environmental conditions (Gilad et al., 2007), which directly affect, for example, water availability, a critical re-
source for plant growth in semi-arid regions (Deblauwe et al., 2008). Therefore, investigating how vegetation systems respond
to different perturbations can provide valuable insights into their resilience and stability.

The seminal works by Holling (1973) established the difference, in ecology, between stability and resilience. The first one
refers to the ability of a system to return to its equilibrium after a perturbation and is linked to the eigenvalues associated

with a given stable equilibrium. Resilience, rather, defines the ability of an ecosystem to maintain its function. For example,
in semi-arid areas, an ecosystem constrained by limited water supply may achieve its highest biomass when vegetation is
distributed in the form of patterns. Furthermore, based on vegetation models, we know that different pattern configurations
may be compatible with a same boundary condition (von Hardenberg et al. (2001), Dijkstra (2011), Zelnik et al. (2013)). As
these different configuration may all achieve high biomass, it is generally conjectured that resilience is effectively enhanced by

the multiplicity of possible patterns.

However, understanding resilience also requires attention to the nature of the perturbation (Kéfi et al., 2019): A system may
react very differently depending the type of perturbation being applied to it.

Our objective is to determine the critical scales, in time and space, which determine the response of the system to environ-
mental perturbations. We consider, first, linear, deterministic perturbations such as a linear decrease in precipitation, and then

stochastic perturbations. We aim establishing general principles. Even though the work is based on a specific numerical model
of vegetation (Rietkerk et al., 2002), we aim at identifying which critical time scales emerge from the model construction, and
how they influence the system's response.

To this end, we follow the framework established by previous studies. More specifically, Siteur et al. (2014), Chen et al.
(2015) and Sherratt (2013) showed the importance of the rate of change and noise and Bastiaansen et al. (2020) identified

in Klaussmeier's vegetation patterns model a critical time scale for which the patterns don't have the time to adapt to the
environmental change. All those works emphasize the importance of understanding rate-induced transitions.

After those studies, a number of questions remain: are there more than one critical time scale in a vegetation pattern model;
what are the impact of those internal timescales, and can they be linked to parameter of the model? In the following we
will show how these critical time scales manifest themselves in Rietkerk's model (Rietkerk et al., 2002), and then show how

they influence both the response to deterministic and stochastic perturbations, with emphasis on the possible occurrence of a
resonance process.

## 2  Model description

As announced in the introduction we use the vegetation model by Rietkerk et al. (2002). As common for reaction-diffusion
models, it combines two mechanisms for creating patterns: local facilitation, and long-range inhibition. Local facilitation is

caused by the water-infiltration feedback. It is based on the idea that in a semi-arid region the soil crust effectively prevents
water infiltration. The presence of biomass and more specifically the roots associated with this vegetation increases the water




infiltration by drilling the soil crust. Long-range inhibition is caused by rapid diffusion of surface water preventing the accumulation of water in some places of the spatial domain and so the creation of biomass. The presence of these two processes places the Rietkerk's model in the category of scale-dependent feedback model. (Lefever and Lejeune (1997), Rietkerk and van de

Koppel (2008), van de Koppel et al. (2005)). Although models of this class are known to produce regular patterns as stable equilibria, Vanderveken et al. (2023) showed that non regular equilibria (Mixed state) exist and play a role in the dynamic of the system in spite of being unstable. Another class of pattern-formation models have been proposed recently by Siteur et al. (2023). This other class of models is based on density-dependent-aggregation of biotic or abiotic species and can create non regular patterns.

Three variables are modelled in Rietkerk's model: Biomass ($B$) [g.m$^{-2}$], soil water ($W$) [mm] and surface water ($O$) [mm]. They respond to to following system of partial differential equations:

$$\frac{\partial B}{\partial t} = cg_{max}\frac{W\,B}{W+k_1} - dB + D_B\Delta B,$$
$$\frac{\partial W}{\partial t} = \alpha O\frac{B+k_2 w_0}{B+k_2} - g_{max}\frac{W\,B}{W+k_1} - r_w W + D_W\Delta W,$$
$$\frac{\partial O}{\partial t} = R - \alpha O\frac{B+k_2 w_0}{B+k_2} + D_O\Delta O, \tag{1}$$

where $\Delta$ is the Laplacian operator and $R$ is the rainfall [mm.d$^{-1}$]. The rainfall is the external forcing of the system which we consider to be a spatially-independent function. The first term in the biomass equation represents water uptake by the plant.

The first term in the soil water equation is linked to the infiltration rate of water in the soil that is enhanced by the presence of biomass. The factors in front of the Laplacians ($\Delta B$, $\Delta W$ and $\Delta O$)are the diffusion constants of the different quantities. We consider a periodic domain of size $l = 100$m. The parameters are given in appendix A.

## 3 Effect of Rainfall Perturbation on Vegetation Dynamics: Identifying Critical Timescales

We consider a spatially homogeneous perturbation applied to surface water, representing a decrease in rainfall across the entire

domain. This type of perturbation is meaningful as it mimics scenarios of prolonged droughts or gradual shifts in climate patterns. By applying different rates of change to a system that is initially in a high rainfall state, we aim to identify which critical factors influence the transient response of vegetation.

Previous research has highlighted the dependence of system solutions on the rate of environmental change, as common in open dynamical systems. In the context of vegetation patterns, different rates of rainfall decrease can lead to diverse outcomes,

ranging from transitions to multi-pulse states to abrupt shifts to bare soil.

Starting from a high rainfall situation ($R = 1.4$mm.day$^{-1}$), for which only the homogeneous vegetated state is stable, we decrease rainfall with different rates of change $a$ ([mm.day$^{-2}$]). Figure 1 shows the evolution of the spatial structure of the biomass (in green) as a function of rainfall, for different values of $a$. We find a transition from an homogeneous state to a heterogeneous distribution with two or three vegetation 'pulses', before it finally disappears. The sequence and timing of the



transitions depend on $a$. For a rate of change of $10^{-3}\,\mathrm{mm.day^{-2}}$ the system jumps directly from a homogeneous vegetated state to a bare soil solution. The critical rate of change at which all vegetation is eradicated is $\tau_{R_{\mathrm{tip}}} \sim 1000$ days.

It is not surprising that the response depends on the rate of the perturbation change. Ashwin et al. (2012) established general principles of so-called rate-induced tipping (R-tipping) in models based on ordinary differential equations, and rate-dependent response were also described specifically in models of vegetation patterns (Siteur et al. (2014), Chen et al. (2015), Bastiaansen

et al. (2020)). However, the value of $\tau_{R_{\mathrm{tip}}}$ is intriguing. Indeed, the timescale associated with the destruction and creation of vegetation pulses is, through dimensional analysis, estimated to be $\tau_{\mathrm{pulse}} = \frac{c}{k_2} \frac{R}{r_w} \frac{1}{d} = 30$days. It is is linked to the transfer of water to the biomass and to the destruction of the vegetation.

Having $\tau_{R_{\mathrm{tip}}}$ about 30 times $\tau_{\mathrm{pulse}}$ suggests that a slower process dominates the system's behavior under changing environmental conditions. Bastiaansen et al. (2020) already identified another critical timescale within the Klausmeier's model

(Klausmeier (1999b)). It is linked to the movement of pulses towards a stable, regular state. This concept is crucial for understanding vegetation dynamics in response to gradual environmental changes. We now proceed to determine a similar time scale in Rietkerk et al. (2002)'s model, and will call it the rearrangement timescale, $\tau_{\mathrm{rear}}$.

First, we perform a numerical experiment where a pulse is removed for a given rainfall value, and the time taken for the system to stabilize is computed. Figure 2 shows the results for different values of rainfall. We find that $\tau_{\mathrm{rear}} \sim 1000$ days. This

rearrangement timescale is of the same order as $\tau_{R_{\mathrm{tip}}}$, suggesting a fundamental connection between the two. This correspondence would also support that R-tipping is conditioned by the slowest timescale in the system. The rearrangement timescale effectively represents this slowest process, dictating the system's overall response to changing conditions. We now attempt to link this rearrangement timescale to quantities in the system. Inspired by the scaling proposed in Bastiaansen and Doelman (2019), we reason on the fact that the movement of pulses are determined by diffusion coefficients. Specifically, we take the

advantage of the fact that the ratio between the slow and the fast diffusion coefficients in the reaction-diffusion model drives the creation of the patterns (Murray (2003),Meron (2015)). For Rietkerk's model the fast component is the surface water ($O$) and the slow components are the biomass ($B$) and the soil water ($W$). Hence, we propose the following scaling for the rearrangement time $\tau_{rear} = \frac{c}{k_2} \frac{R}{r_w} \frac{1}{d} \sqrt{\frac{D_O}{D_B}} = 1000$d. The key factor determining the rearrangement time appears to be the square root of the ratio between the diffusion coefficients of the fast and slow variables. To verify this, we perform a series of experiments

where the diffusion coefficient of the slow variables (biomass and soil water) varied by a factor $f$. To obtain a stabilisation time for the different values of $f$, we run a series of numerical experiments similar to the one presented earlier. We start from a stable state for a given value of rainfall and $f$, then we remove one pulse of vegetation. We compute this stabilisation time for different values of rainfall and took the larger stabilisation time because it is the one that is of interest regarding R-tipping. The results are presented in Figure 2. The rearrangement time scale which is diagnosed numerically fits the theoretical curve

proposed for $\tau_{\mathrm{rear}}$ fairly well.

This finding highlights the critical role of the rearrangement timescale in determining the system's response to environmental changes. Specifically, it emphasizes the need to account for this slower timescale when analyzing vegetation dynamics. Understanding and quantifying this timescale provides valuable insights into the resilience and stability of vegetation systems under gradual environmental shifts.



## 4  Effect of Spatially and Temporally Structured Noise on Vegetation Pattern Resilience

The kind of perturbations applied to the system is an important aspect of resilience diagnosis (Kéfi et al. (2019)) and specifically, the spatial scale of the perturbation must be considered. In the following, we consider a dynamic perturbation consisting of a spatially distributed noise modeled mathematically as a Gaussian stochastic process with mean $\mathbb{E}[\xi(t,x)] = 0$ and standard deviation $\sigma = 0.1$. The temporal structure is that of an Orstein-Uhlenbeck process. The correlation function is $\mathbb{E}[\xi(t,x)\xi(s,y)] = e^{-|t-s|/\lambda_t}e^{-(x-y)^2/\lambda_s^2}$.

Stochastic perturbations are applied to the biomass $B$ and surface water $W$. Formally, the model takes the form:

$$\frac{\partial B}{\partial t} = cg_{max}\frac{WB}{W+k_1} - dB + D_B\Delta B + \xi_B(x,t),$$
$$\frac{\partial W}{\partial t} = \alpha O\frac{B+k_2w_0}{B+k_2} - g_{max}\frac{WB}{W+k_1} - r_wW + D_W\Delta W,$$
$$\frac{\partial O}{\partial t} = R - \alpha O\frac{B+k_2w_0}{B+k_2} + D_O\Delta O + \xi_O(x,t). \tag{2}$$

where $\xi_O(x,t)$ and $\xi_B(x,t)$ are independent noise processes. For biomass, we consider an autocorrelation timescale of one day, and different autocorrelation length-scales ($\lambda_{biomass,s}$) are tested. For surface water, the perturbation is homogeneous in space, in different autocorrelation time-scales $\lambda_{rainfall,t}$ are tested.

Figure 4 summarises the stationary response of the model to $\lambda_{biomass,s}$ and $\lambda_{rainfall,t}$, considering the mean biomass. Outputs reported on those tables is the average of 15 experiments. First, consider the spatial structure of the noise, applied to the biomass. That is, we focus on the horizontal lines of the tables, one by one, with the structure of the perturbation changing from highly heterogeneous to almost homogeneous as each line is browsed from the left to the right (increasing $\lambda_{biomass,s}$) We find that mean biomass tends to decrease as spatial autocorrelation increases, and this occurs regardless the choice of time autocorrelation. This suggests that more homogeneous stochastic perturbations tend to destroy more biomass and send the system towards the bare soil equilibrium. The resilience of the vegetation is thus reduced if the perturbation is spatially more homogeneous.

We now focus on the time-correlation of the stochastic perturbation of surface water. We find that both the spatial mean *and* the variance reach a minimum at $\lambda_t = 10$d. This timescale is the $\tau_{\text{pulse}}$ timescale previously identified by the scale analysis. This suggests a form of resonance associated with $\tau_{\text{pulse}}$ preventing self-organisation of vegetation. On the other hand, biomass reaches a maximum for a perturbation time scale of 1000 days, coinciding with the rearrangement timescale $\tau_{\text{rear}}$ that we also identified by scaling analysis.

To visualise the behavior for those different values of temporal autocorrelation, we show, in Figure 5, realisations representative of the general behavior, for four values of $\lambda_t$. We see that patterns tend to disappear for $\lambda_t = 10$d and reappear for $\lambda_t = 1000$d. The ability of the system to maintain or not patterns, depending on the time scale of the pertrubation, explains the variation of mean biomass observed in Figure 4. We suggest that the slow stochastic perturbation is more effective at moving the system gently around the states associated with the highest biomass, without destroying patterns. The slow timescale (rearrangement timescale) is intrinsically linked to the spatial extension of the model. It would not exist in a zero-dimensional





analysis. In this respect, we have noted already that non-linear, spatially extended systems tend to have multiple of equilibria for a given input, here, a given rainfall. The multiplicity of equilibria, whether they are stable or not, increases the resilience of the system in the sense of Holling (1973). Metaphorically, such equilibria may be seen as tree branches in a forest to which an orangutan might cling to avoid falling. This phenomenon is clearly visible on Figure 5, displaying the time evolution of the biomass with a fixed rain fall of $0.8 \text{mmd}^{-1}$. In this configuration, we see that the systems jumps from a pulse configuration to

an other, passing from two to three pulses or the other way around. The transition, associated with the vanishing or the creation of a pulse, happens quickly, with a time scales of a few days. We find again the $\tau_{\text{pulse}}$ (fast) timescale identified earlier. Those configurations correspond to stable equilibria for the chosen rainfall as previously identified (Zelnik et al. (2013), Vanderveken et al. (2023)). The two and three pulse configurations are not the only states being visited by the system. From time to time a one big pulse with a small one pulse configuration appears. The latter does not correspond to an equilibrium for a rainfall value

of $0.8 \text{mmd}^{-1}$, but it can be interpreted as the ghost of a *mixed state* identified in Vanderveken et al. (2023). Mixed states are always unstable and exhibit pulses with different heights. The mixed state that interests us in here is the one with one big pulse and one small pulse. Its existence spans from $0.6 \text{mmd}^{-1}$ to less than $0.8 \text{mmd}^{-1}$. In the stochastic realisation, the position of the second small pulse is not perfect because of noise. We clearly see that even if this mixed state with two pulses is not a equilibria for this value of rainfall, it plays a role in the dynamics, hence the qualifier of "ghost" (Hastings et al. (2018) and

Morozov et al. (2020)). Ghost attractors and unstable equilibria expand the resilience of the system because, in view of the orangutan in the forest, those configuration are new tree branch to cling to.

By demonstrating that resilience is bolstered by spatially heterogeneous noise and identifying critical timescales that influence biomass dynamics, we have provided valuable insights into the mechanisms underpinning ecological stability. Our findings emphasize the necessity of considering both stable and metastable states in resilience assessments, offering a more

comprehensive understanding of ecosystem dynamics under varying perturbation regimes.

## 5   Conclusions

The present study explored the effects of a spatially homogeneous perturbation, specifically a decrease in rainfall, on vegetation dynamics in a model system. By varying the rate of change in rainfall, we observed significant differences in the system's response. At slower rates, the system transitions through multiple stable states, including two and three pulse solutions. However,

a critical rate of change at $10^{-3}, \text{mm.d}^{-2}$ results in a direct shift from a homogeneous vegetated state to bare soil. This critical rate of change aligns with a critical timescale $\tau_{R_{\text{tip}}} \sim 1000$ d, much longer than the pulse creation and destruction timescale $\tau_{\text{pulse}} = 30 \text{d}$, highlighting the dominance of slower processes in the system's behavior under environmental changes.

### 5.1   Rearrangement Timescale and System Resilience

We identified a critical rearrangement timescale $\tau_{\text{rear}} \sim 1000$ d, which corresponds to the system's response time to structural

changes, such as the removal of a vegetation pulse. This timescale is pivotal for understanding the resilience and stability



of vegetation patterns. By linking $\tau_{\text{rear}}$ to the diffusion coefficients of the system's components, we derived a scaling law that accurately predicts the rearrangement time, emphasizing the importance of spatial interactions in determining system dynamics.

## 5.2 Impact of Spatial and Temporal Noise

We further investigated the resilience of the system under spatial and temporal noise, modeled as Gaussian stochastic processes.
The experiments demonstrated that the spatial autocorrelation of noise significantly impacts pattern formation. Higher spatial correlation lead to reduced spatial variance and biomass, suggesting that more homogeneous noise disrupts vegetation patterns. Temporal noise analysis revealed critical timescales—particularly around 10 days and 1000 days—corresponding to the pulse and rearrangement timescales, respectively. These findings underscore the intricate balance between spatial and temporal scales in maintaining vegetation resilience.

## 5.3 Broader Implications and Multiple Equilibria

Our analysis highlights the existence of multiple equilibria and the crucial role of slow timescales in spatially extended systems. The presence of ghost attractors and unstable equilibria broadens the system's resilience, offering additional configurations for the system to cling to, akin to tree branches in a forest. This multiplicity of stable and transient states enhances the system's capacity to adapt to gradual environmental changes, aligning with the concept of resilience as defined by Holling (1973).

In summary, this study contributes to elucidate a complex interplay between rainfall perturbations, spatial and temporal noise, and vegetation dynamics. By identifying critical timescales and exploring the system's response to various perturbations, we provide valuable insights into the resilience and stability of vegetation systems. These findings contribute to a deeper understanding of ecological dynamics and the factors influencing the persistence of vegetation patterns under changing environmental conditions.

*Code availability.*   All the code used to produce the figures for this paper is available here: https://zenodo.org/doi/10.5281/zenodo.13739303 (Vanderveken, 2024)

## Appendix A:  Model parameters

The parameters are as in Rietkerk et al. (2002), see table A1

## Appendix B:  Structured noise

The noise used in this paper is correlated both in time and space. We prescribe its standard deviation $\sigma$, length of spatial autocorrelation $\lambda_s$ and time correlation $\lambda_t$ with a period of $L$. To produce this noise, first, we computed the square root of the



| $c$ | Conversion of water uptake by plants to plant growth | $10\,\mathrm{g\,mm^{-1}\,m^{-2}}$ |
|---|---|---|
| $g_{max}$ | Maximum water uptake | $0.05\,\mathrm{mm\,g^{-1}\,m^{-2}\,d^{-1}}$ |
| $k_1$ | Half-saturation constant of specific plant growth and water uptake | $5\,\mathrm{mm}$ |
| $D_B$ | Plant dispersal | $0.1\,\mathrm{m^2\,d^{-1}}$ |
| $\alpha$ | Maximum infiltration rate | $0.2\,\mathrm{d^{-1}}$ |
| $k_2$ | Saturation constant of water infiltration | $5\,\mathrm{g\,m^{-2}}$ |
| $w_0$ | Water infiltration in the absence of plants | $0.2$ |
| $r_w$ | Soil water loss due to evaporation and drainage | $0.2\,\mathrm{d^{-1}}$ |
| $D_W$ | Diffusion coefficient for soil water | $0.1\,\mathrm{m^2\,d^{-1}}$ |
| $D_O$ | Diffusion coefficient for surface water | $100\,\mathrm{m^2\,d^{-1}}$ |
| $d$ | Plant mortality rate | $0.25\,\mathrm{d^{-1}}$ |

**Table A1.** Parameters for Rietkerk's model

following covariance matrix where $N = 100$ is the number of spatial points.

$$\mathbf{A}_{i,j} = max\left( e^{\frac{-(x_i - x_j)^2}{\lambda_s^2}}, e^{\frac{-(L - |x_i - x_j|)^2}{\lambda_s^2}} \right)$$

Then we use an AR1 process in time to create the time dependence of the noise

$$x_{i,j} = x_{i,j} + (-c * x_{i,j} dt + \sqrt{dt}\sigma_e \sum_{k=1}^{n} \mathbf{A}_{j,k}\epsilon_k)$$

with $c = 1/\lambda_t$, $dt$ the time step, $\sigma_e = \sigma\sqrt{2c}$ and $\epsilon_k$ a Gaussian random variable of zero mean and variance one.

The resulting noise is exemplified if A1 and the temporal and spatial structure are shown in A2.

*Author contributions.* LV and MC designed the study. LV performed the numerical analysis. LV wrote the paper, and MC reviewed and edited the paper

*Acknowledgements.* The authors used chat-GPT3.5 to help rephrase some parts of the paper.





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



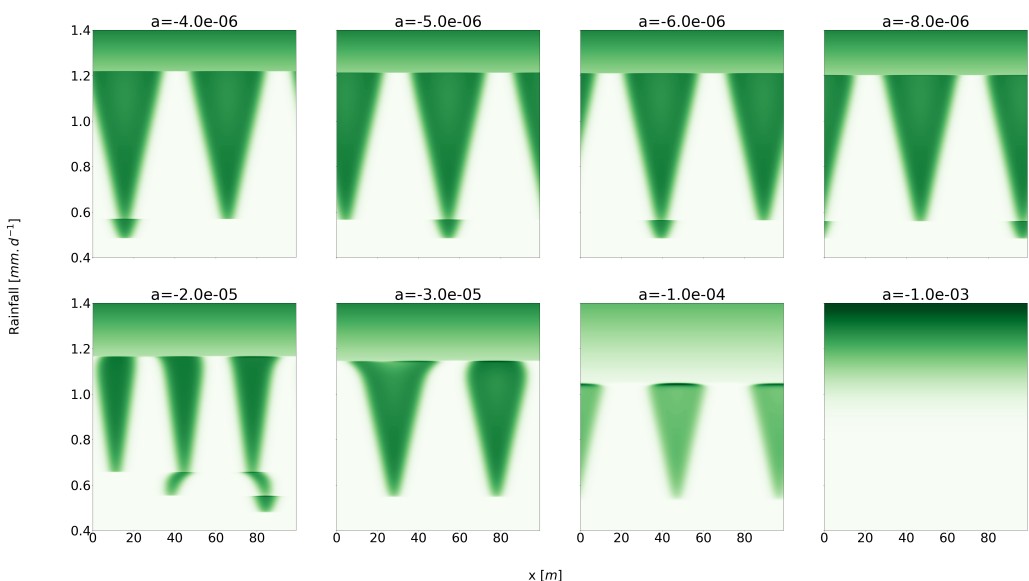

**Figure 1.** Sensitivity of Rietkerk's model to various rates of change in the rainfall. Each panel represents a solution with a given rate of change $a$. The x-axis is the spatial dimension, and y-axis the rainfall. Every run has the same starting and ending point rainfall values but they differ by the rate of change. Every simulation starts with a homogeneous vegetated state which is the only stable equilibrium at this value of rainfall. For the last value of rainfall ($R = 0.4\text{mm.days}^{-1}$), the only stable equilibrium is the homogeneous bare soil solution.





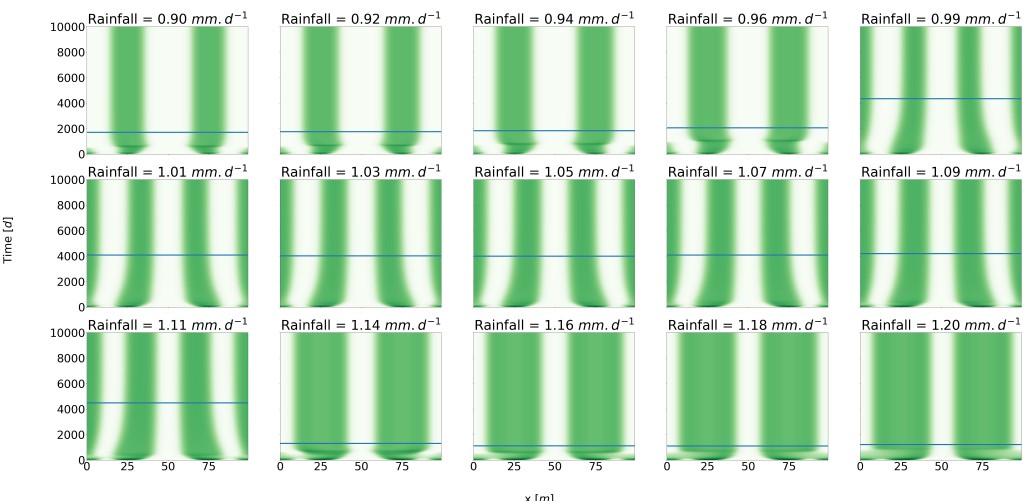

**Figure 2.** Rearrangement time scale with respect to the rainfall. Each panel represents the solution of the following numerical experiments, we start in the stable equilibrium with four pulses and we remove one pulse. We then identify for each experiment the time until the system reaches a new equilibrium by moving around the pulses. This time is shown by a horizontal blue line on each panel.



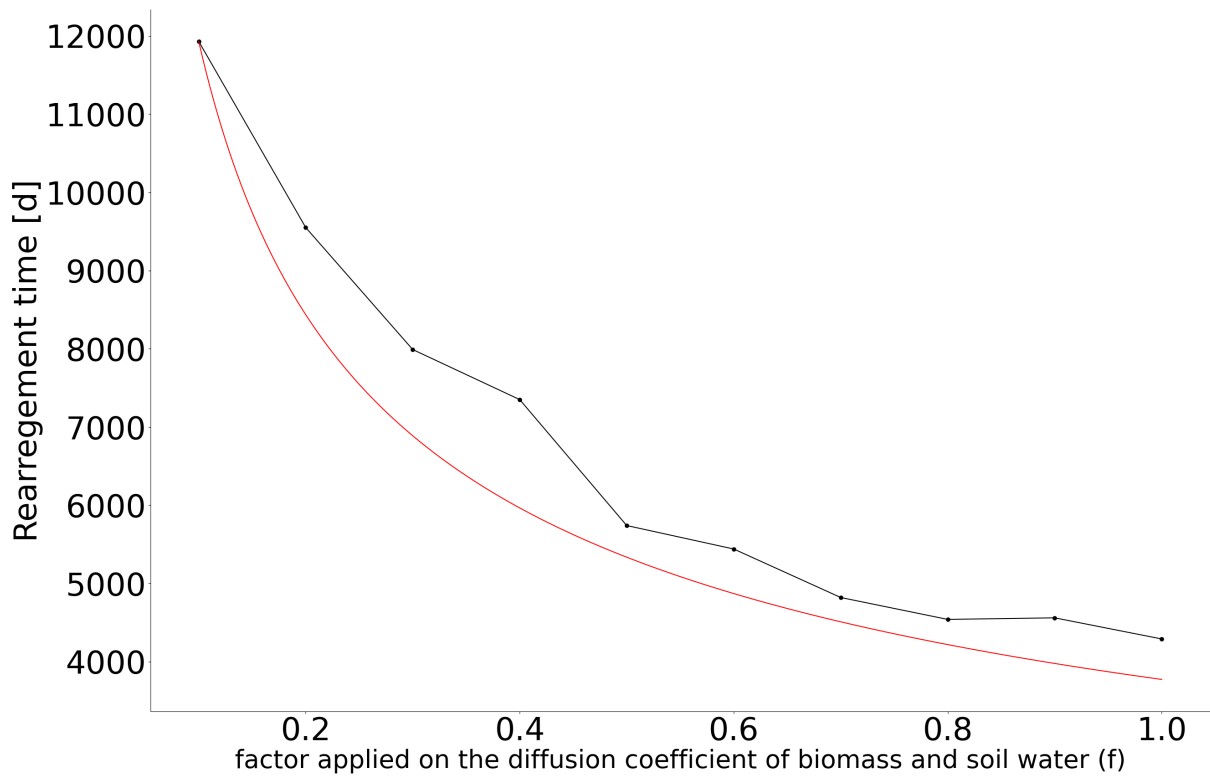

**Figure 3.** Link between the factor $f$ applied on the diffusion coefficient of the the "slow" components (biomass and soil water) and the rearrangement timescale. This timescale is estimated by finding a stable equilibrium for the new diffusion coefficients, remove one of the pulse and then estimate the stabilisation time. This procedure is applied for different values of rainfall and we take into account the largest stabilisation time. The results are the black dot and the red line scales as $\frac{1}{\sqrt{x}}$



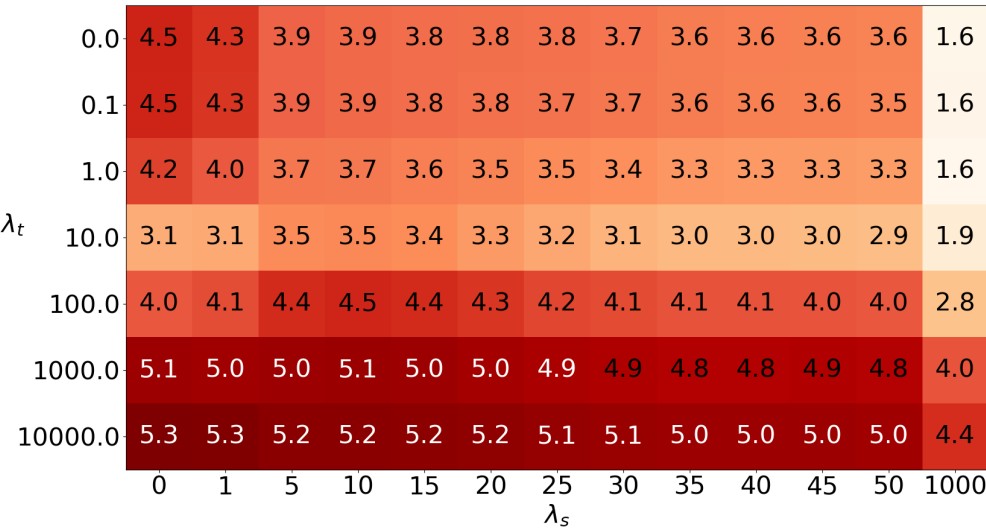

**Figure 4.** Summary table for the runs with stochastic forcing. For each cell, we ran the model 15 times with the same $\lambda_t$ and $\lambda_s$. The table indicates the ensemble mean of the temporal mean of the spatial mean.



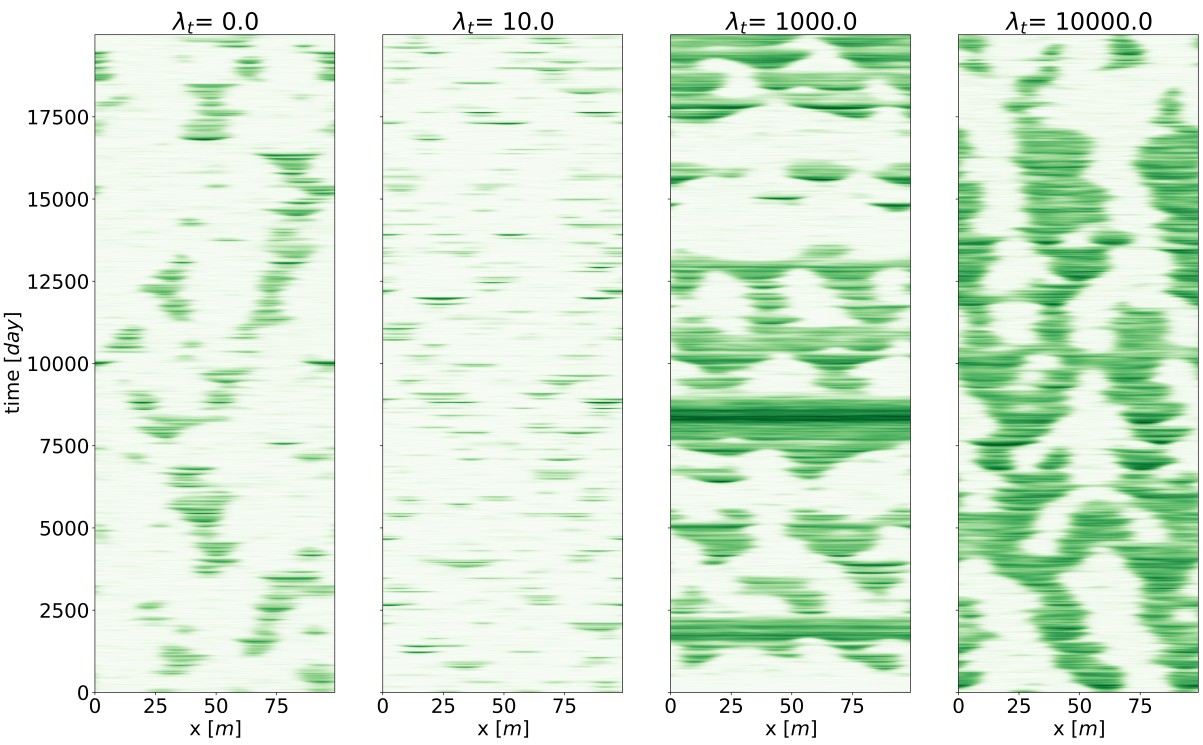

**Figure 5.** Four realisations of the stochastic Rietkerk model with $\lambda_s = 1$m. The biomass is the variable shown. Each panel shows a representative realisation with different temporal autocorrelations ($\lambda_t$).

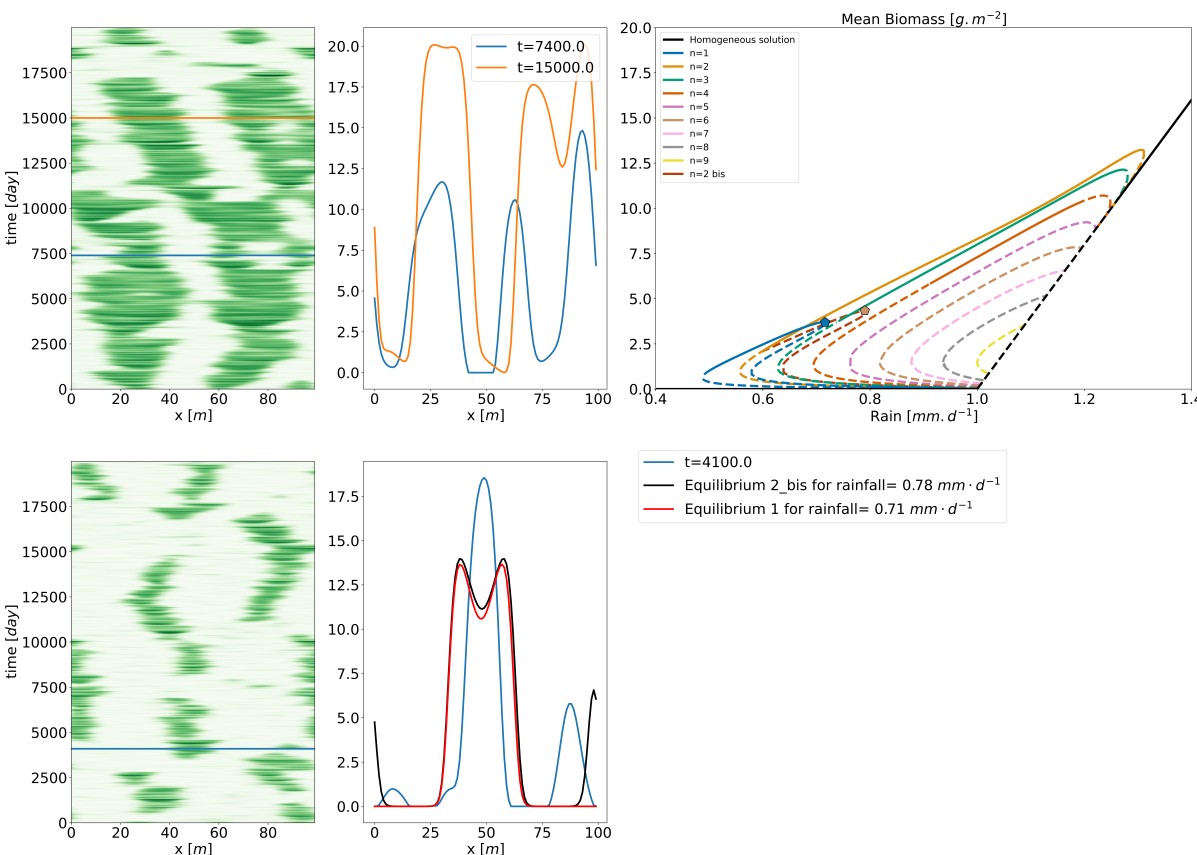

**Figure 6.** Two realisations of the stochastic Rietkerk model with different noises. On the top-left panel, the parameters are $R = 0.8\text{mmd}^{-1}$, $\lambda_t = 1\text{d}$ and $\lambda_s = 10\text{m}$ for the noise applied on the biomass, and $\lambda_t = 10000\text{d}$ and spatially homogeneous for the noise applied on the surface water. On the top-bottom panel, the parameters are $R = 0.8\text{mmd}^{-1}$, $\lambda_t = 1\text{d}$ and $\lambda_s = 15\text{m}$ for the noise applied on the biomass, and $\lambda_t = 1\text{d}$ and spatially homogeneous for the noise applied on the surface water. The three horizontal lines mark the timing of the snapshots represented on the right panels respectively. The top-right panel gives the bifurcation diagram with two pentagons marking the position of two equilibria shown on the bottom-center panel.



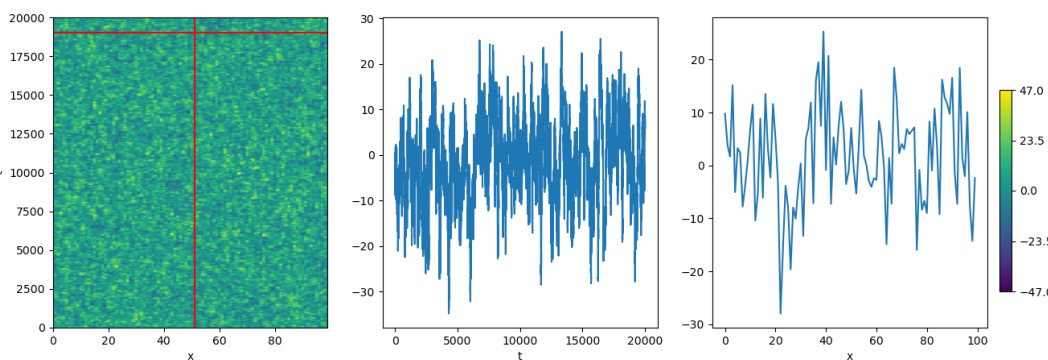

**Figure A1.** One realisation of the structured noise with $\sigma = 10$, $\lambda_t = 100$ and $\lambda_s = 1$. On the left panel is shown the full noise. The vertical red line and the horizontal red line mark the position of the snapshots, those snapshots are represented on the middle panel for the vertical one, and on the left panel for the horizontal one.



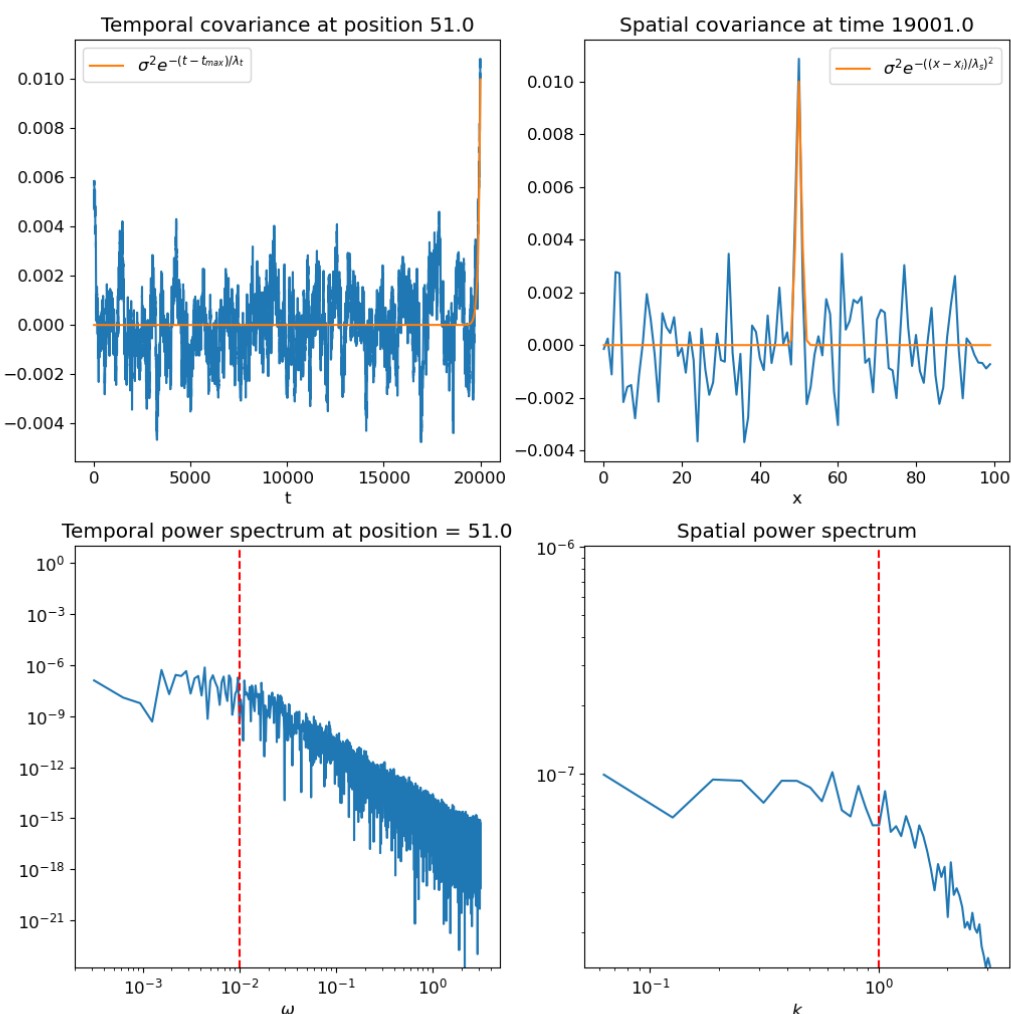

**Figure A2.** Mean over 50 realisations of the structured noise with $\sigma = 10$, $\lambda_t = 100$ and $\lambda_s = 1$. One the upper-left panel, the blue line is the mean over all the realisations with temporal autocorrelation at $t = 20000$; the orange line is the theoretical expectation. One the upper-right panel, the blue line is the mean over all the realisations of the spatial autocorrelation at $x = 50$, the orange line is the theoretical expectation. On the bottom-left panel the blue line is the mean temporal power spectrum, and the vertical red dashed line is at $1/\lambda_t$. On the bottom-left panel the blue line is the mean spatial power spectrum and the vertical red dashed line is at $1/\lambda_s$.