# Peer review of "Rate-Induced Transitions and Noise-Driven Resilience in Vegetation"

_EGUsphere, 2024_

## Author Response (AR1)

**Response to the reviewer**

**Lilian Vanderveken, Michel Crucifix**

**Response to Reviewer #1**

We appreciate your thorough review of our manuscript and the constructive feedback provided. Below, we have addressed each of your comments in detail, describing the changes made to the manuscript where applicable.

**Q1: It would be useful to assess the stability of the results with respect to the intensity of the noise. I mean finding, if it exists, a relation and how it is robust between the critical transitions with varying noise amplitude. This would provide more information on the assessment of the role of noise-driven processes.**

To address the stability of the results with respect to the intensity of noise and explore the relationship between critical transitions and varying noise amplitude, we conducted additional numerical experiments. These experiments assess the role of noise amplitude ($\sigma$) on the system dynamics and the robustness of observed transitions. We considered three noise levels: low ($\sigma = 0.09$), intermediate ($\sigma = 0.3$), and high ($\sigma = 0.9$).

For low noise intensity ($\sigma = 0.09$), the results indicate that the observed dynamics remain consistent with those published, suggesting that the system retains its structural features.

[Figure]

Figure 1: Summary table for the runs with stochastic forcing ($\sigma = 0.09$). For each cell, we ran the model 15 times with the same $\lambda_t$ and $\lambda_s$. The table indicates the ensemble mean of the temporal mean of the spatial mean.

At intermediate noise intensity ($\sigma = 0.3$), the patterns are destroyed at a temporal autocorrelation scale of $\lambda_t = 1$ d, but they reappear at higher temporal autocorrelation values.

[Figure]

Figure 2: Five realizations of the stochastic Rietkerk model with $\lambda_s = 5$m. The biomass is the variable shown. Each panel shows a representative realization with different temporal autocorrelations ($\lambda_t$) for $\sigma = 0.3$.

At high noise intensity ($\sigma = 0.9$), patterns are completely destroyed, and the system stabilizes into a spatially homogeneous state that varies with rainfall input. This indicates a saturation effect where noise dominates the system dynamics, overriding any spatial pattern formation processes.

Regarding mean biomass, we observed for both $\sigma = 0.3$ and $\sigma = 0.9$, a consistent increase with increasing temporal autocorrelation.

[Figure]

Figure 3: Summary table for the runs with stochastic forcing ($\sigma = 0.3$).

We further analyzed the influence of noise intensity on the mean amount of biomass with two complementary experiments. First, we apply noise to the surface water only. For low noise amplitude ($\sigma = 0.1$), biomass is eliminated around the critical time scale $\lambda_t = 10$d. For

[Figure]

Figure 4: Summary table for the runs with stochastic forcing ($\sigma = 0.9$).

higher noise amplitudes ($\sigma = 0.3$ and $\sigma = 0.9$), two effects emerge: (1) the range of temporal autocorrelation leading to biomass reduction shifts to lower $\lambda_t$ values, and (2) mean biomass increases significantly at higher $\lambda_t$. Together, these effects demonstrate an increasing trend in mean biomass with greater temporal autocorrelation at higher noise intensities.

[Figure]

Figure 5: Temporal mean of the spatial mean of the biomass with respect to temporal auto-correlation with noise applied on surface water

Second, when noise is applied exclusively to biomass, with a fixed temporal autocorrelation ($\lambda_t$=1 d), we observe that mean biomass increases consistently with noise amplitude. This can be attributed to the asymmetrical of the effects of noise: positive noise contributions benefit biomass more than negative noise reduce it, which we interpret as a consequence of the non-negativity constraint of biomass. This is pronounced as noise amplitude increases, leading to an overall increase in biomass.

These results demonstrate that phenomena associated with the destruction and reappearance of patterns are robust to moderate noise levels ($\sigma = 0.3$) but are suppressed entirely at high

[Figure]

Figure 6: Temporal mean of the spatial mean of the biomass with respect to spatial autocorrelation with noise applied on biomass

noise intensities ($\sigma = 0.9$), where the system becomes fully noise-driven. The observed increase in mean biomass with increasing temporal autocorrelation reflects the interplay of noise effects on both surface water and biomass components,

We add this discussion to the manuscript in the following way:

line 168-188:

*In this study, we focused on the effect of spatial and temporal correlations on the system. To assess the impact of noise amplitude, we performed numerical experiments for low noise intensity $\sigma = 0.09$ and high noise intensity levels, $\sigma = 0.3$ and $\sigma = 0.9$. We observed that the results are consistent for low value of noise amplitude.*

*For $\sigma = 0.3$, we observe pattern destruction at $\lambda_t = 1d$, followed by pattern reappearance at higher values of temporal autocorrelation. At $\sigma = 0.9$, all patterns are destroyed, and the system stabilizes into a spatially homogeneous solution oscillating with rainfall.*

*Regarding mean biomass, we observed for both $\sigma = 0.3$ and $\sigma = 0.9$, a consistent increase with increasing temporal autocorrelation. This behaviour can be understood by two complementary numerical experiments.*

*First, we applied noise exclusively to the surface water, using three values of $\sigma = 0.1, 0.3, and 0.9$. For $\sigma = 0.1$, biomass is eliminated at $\lambda_t = 10d$. For higher noise amplitudes, two phenomena emerge: (1) the range of temporal autocorrelation over which biomass is reduced shifts to lower values of $\lambda_t$, and (2) the mean biomass increases significantly at higher $\lambda_t$. Together, these effects result in an increase in biomass with increasing temporal autocorrelation.*

*Second, we applied noise exclusively to the biomass components, using the same three values of $\sigma$ and fixing temporal autocorrelation at $\lambda_t = 1d$. In this setup, we observe that mean biomass*

*increases with σ. This can be attributed to the nature of the Gaussian noise applied. Gaussian noise is symmetric, and since biomass is constrained to be non-negative, the system benefits more from the positive phases of the noise while being less affected by the negative phases. As the amplitude of the noise increases, this asymmetry leads to an overall increase in biomass.*

*The combined effects of noise applied to both surface water and biomass explain the observed increase in mean biomass for σ > 0.1. However, the destruction and reappearance of patterns remain robust even at higher noise amplitudes (e.g., σ = 0.3). At sufficiently high noise intensity (σ = 0.9), a saturation effect occurs, and the system becomes entirely noise-driven, with patterns fully suppressed.*

**Q2: I find the results in Figure A2 very interesting and maybe suitable for description after considering my previous point. Indeed, it would be interesting to show directly a k-ω spectrum as a function of the noise intensity and how temporal/spatial covariance is captured through the noise process.**

Figure A2 illustrates the noise applied to the system, which was generated using an Ornstein-Uhlenbeck process with a defined temporal autocorrelation and a spatial structure based on a covariance matrix for a periodic domain. This process ensures independence between temporal and spatial structures.

Given the independence of these structures and that a k-ω spectrum is more suitable for analyzing noise where temporal and spatial structures are entangled. the k-ω spectrum does not add information to the analysis. Regarding the impact of the noise intensity, since the noise generation process is linear, increasing the standard deviation (σ) does not alter the noise characteristics.

**Q3: My last point is on Figure 3. Would be possible to make an exponential fit of the black line to be compared with the resulting scaling?**

The purpose of Figure 3 was to confirm, via numerical experiments, the relationship between rearrangement timescales and the diffusion coefficients of biomass and soil water, which follow a $1/\sqrt{x}$ scaling.

While an exponential fit produces better results due to the additional parameter, we believe that it does not enhance our understanding of the relationship between rearrangement timescales and model parameters.

**Q4: As a minor point I would suggest a careful reading of the text to fix some typos, as below:** Thank you for popinting those typos. We performed a typo check on the whole text

**Response to reviewer #2**

We are grateful for your valuable feedback, which has helped improve the clarity of our manuscript. Below, we address your comments in detail.

[Figure]

Figure 7: Relation between the rearrengement timescale and the values of the diffusion constants of biomass and soil water. The red line represents a $1/\sqrt{x}$ and the blue line is the exponential fit

**Q1: The responses of vegetation to reduced rainfall rates is discussed in the manuscript. How about the responses in an increased rainfall scenario? Do these two inverse scenarios induce anti-symmetric responses? In other words, are the changes of vegetations due to increased and decreased rainfall reversible?**

We performed reverse numerical experiments to explore the system's response to increased rainfall. Due to the system's multistability, hysteresis behavior is observed.

[Figure]

We add a comment in that in the manuscript

Line 85-84: *The existence of multistability implies hysteresis behaviour. When reversing the*

*rainfall gradient—from low to high—the system takes a different paths for the transition between low and high biomass.*

**Q2: Noises are applied to two variables— the biomass B and surface water W. What if the noises are applied to only one variables or all three variables?**

In the original setup, noise was applied uniformly to the surface water variable to represent spatially uniform convective rainfall events, considering a domain size of 100 m. For biomass, the noise's temporal autocorrelation is set to one day to simulate the effects of grazing. While applying noise with smaller spatial structures to surface water or larger temporal autocorrelation to biomass may not be entirely realistic, we introduced structured noise to both biomass and surface water to evaluate its impact. The results are summarized in the tables below.

[Figure]

Figure 8: Summary table of biomass for runs with noise applied on Surface water ($\sigma = 0.1$). For each cell, we ran the model 15 times with the same $\lambda_t$ and $\lambda_s$. The table indicates the ensemble mean of the temporal mean of the spatial mean.

When noise is applied to surface water (representing precipitation variation), the behaviour observed is consistent with the original setup.

However, applying noise to biomass introduced deviations, particularly in the temporal autocorrelation at which pattern destruction occurs across different spatial autocorrelation levels. This shift in the destructive temporal autocorrelation, compared to the initial case, arises because the pulse creation and destruction timescale ($\tau_{pulse}$) identified in the paper depends on rainfall and the rate of water uptake by plants. As a result, the effects of biomass noise differ from those observed in the original setup.

Thank you for pointing out the typos, they are now fixed in the manuscript.

[Figure]

Figure 9: Summary table of biomass for runes with noise applied on Biomass ($\sigma = 0.1$). For each cell, we ran the model 15 times with the same $\lambda_t$ and $\lambda_s$. The table indicates the ensemble mean of the temporal mean of the spatial mean.

---

## Author Response (AR2)

**Post-review modification**

**Lilian Vanderveken and Michel Crucifix**

Dear editor,

Thank you for allowing us the opportunity to make modifications to our manuscript before publication.

The revision concerns the scaling relationship between rainfall and the destruction timescale of vegetation pulses in Section 3, Effect of Rainfall Perturbation on Vegetation Dynamics: Identifying Critical Timescales.

In the original manuscript, we proposed a single timescale ($\tau_{pulse}$) for both the creation and destruction of vegetation patterns, assuming a linear dependence on rainfall based on dimensional analysis. However, after discussions with my thesis jury, we decided to refine this approach. We now define only the destruction timescale, corresponding to the timing of vegetation pulse collapse. A new dimensional analysis led us to a revised scaling law, showing that the destruction timescale scales cubically with rainfall:

$$\tau_{\text{destruction}} = \frac{\alpha}{c\,g_{max}} \left( \frac{R}{k_2\,g_{max}} \right)^3 \frac{1}{d} = 52\text{d}$$

While the order of magnitude remains consistent with our previous estimate, this cubic dependence was further validated through numerical experiments.

To test the proposed scaling, we conducted additional simulations, where we started from a stable equilibrium consisting of two vegetation pulses at $R = 1.2$ mm/d and abruptly reduced rainfall to values between 0.3 mm/d and 0.8 mm/d. This perturbation led to vegetation collapse, and we determined the destruction timescale by fitting an exponential function to the mean biomass evolution. The results, presented in the right panel of Fig. 1, confirm the cubic relationship between rainfall and the destruction timescale.

We have updated the manuscript to reflect these refinements. The core results and interpretations remain unchanged, but we have revised the text accordingly and removed references to the creation timescale. Additionally, we refined some statements in the discussion to reflect the corrected scaling law. We also modified the rearrengement timescale accordingly to reflect this new scaling.

[Figure]

Figure 1: The left panel shows the mean biomass over time for rainfall values ranging from $0.3 \text{mm.d}^{-1}$ to $0.8 \text{mm.d}^{-1}$. The right panel displays the destruction timescales obtained from an exponential fit as a function of rainfall, along with polynomial fits of the destruction timescale of orders one, two, and three.

*line 90-100:*

It is not surprising that the response depends on the rate of change. Ashwin et al. (2012) established general principles of so-called rate-induced tipping (R-tipping) in models based on ordinary differential equations, and rate-dependent responses were also described specifically in models of vegetation patterns (Siteur et al. (2014), Chen et al. (2015), R. Bastiaansen et al. (2020)). However, the value of $\tau_{R_{\text{tip}}}$ is intriguing. Indeed, the timescale associated with the destruction of vegetation pulses is, through dimensional analysis, estimated to be $\tau_{\text{destruction}} = \frac{\alpha}{c\,g_{max}} \left( \frac{R}{k_2\,g_{max}} \right)^3 \frac{1}{d} = 52 \text{d}.$

This relationship is linked to the transfer of water to biomass, to vegetation mortality and to rainfall. To test the dependence of the proposed scaling on rainfall, we conducted the following numerical experiments. Starting from a stable equilibrium consisting of two vegetation pulses at $R = 1.2 \text{mm.d}^{-1}$, we reduced abruptly the rainfall to values between $0.3 \text{mm.d}^{-1}$ and $0.8 \text{mm.d}^{-1}$. This reduction leads to vegetation collapse, driving system toward the bare soil equilibrium. We then determined the destruction timescale by fitting an exponential function to the mean biomass evolution for different rainfall values. The resulting timescales are shown in the right panel of Fig. 1. Our results reveal a cubic relationship between the destruction timescale and rainfall, supporting the validity of our scaling.

*line 110- 115:*

Inspired by the scaling proposed in Robbin Bastiaansen and Doelman (2019), we reason on the fact that the movement of pulses are determined by diffusion

**coefficients. Specifically, we take the advantage of the fact that the ratio between the slow and the fast diffusion coefficients in the reaction-diffusion model drives the creation of the patterns (Murray (2003),Meron (2015)). For the Rietkerk model the fast component is the surface water ($O$) and the slow components are the biomass ($B$) and the soil water ($W$). Hence, we propose the following scaling for the rearrangement time $\tau_{rear} = \frac{\alpha}{c\,g_{max}} \left( \frac{R}{k_2\,g_{max}} \right)^3 \frac{1}{d} \sqrt{\frac{D_O}{D_B}} \sim 1000$d.**

To support these clarifications, we have also included a new figure illustrating the updated scaling relationship. Since these modifications primarily concern explanatory text and do not alter the main findings, we believe they improve the clarity and correctness of the manuscript.

**References**

Ashwin, Peter, Sebastian Wieczorek, Renato Vitolo, and Peter Cox. 2012. "Tipping points in open systems: Bifurcation, noise-induced and rate-dependent examples in the climate system." *Philosophical Transactions of the Royal Society A: Mathematical, Physical and Engineering Sciences* 370 (1962): 1166–84. https://doi.org/10.1098/rsta.2011.0306.

Bastiaansen, R., A. Doelman, Maarten B. Eppinga, and M. Rietkerk. 2020. "The effect of climate change on the resilience of ecosystems with adaptive spatial pattern formation." *Ecology Letters* 23 (3): 414–29. https://doi.org/10.1111/ele.13449.

Bastiaansen, Robbin, and Arjen Doelman. 2019. "The dynamics of disappearing pulses in a singularly perturbed reaction–diffusion system with parameters that vary in time and space." *Physica D: Nonlinear Phenomena* 388: 45–72. https://doi.org/10.1016/j.physd.2018.09.003.

Chen, Yuxin, Theodore Kolokolnikov, Justin Tzou, and Chunyi Gai. 2015. "Patterned vegetation, tipping points, and the rate of climate change." *European Journal of Applied Mathematics* 26 (6): 945–58. https://doi.org/10.1017/S0956792515000261.

Meron, Ehud. 2015. *Nonlinear physics of ecosystems.* CRC Press. https://doi.org/10.1201/b18360.

Murray, James D. 2003. *Mathematical Biology II : Spatial Models and Biomedical Applications , Third Edition.* Springer New York, NY.

Siteur, K., E. Siero, Maarten B. Eppinga, Jens D. M. Rademacher, A. Doelman, and M. Rietkerk. 2014. "Beyond Turing: The response of patterned ecosystems to environmental change." *Ecological Complexity.* https://doi.org/10.1016/j.ecocom.2014.09.002.